# MiRNA-Mediated Fibrosis in the Out-of-Target Heart following Partial-Body Irradiation

**DOI:** 10.3390/cancers14143463

**Published:** 2022-07-16

**Authors:** Barbara Tanno, Flavia Novelli, Simona Leonardi, Caterina Merla, Gabriele Babini, Paola Giardullo, Munira Kadhim, Damien Traynor, Dinesh K. R. Medipally, Aidan D. Meade, Fiona M. Lyng, Soile Tapio, Luca Marchetti, Anna Saran, Simonetta Pazzaglia, Mariateresa Mancuso

**Affiliations:** 1Laboratory of Biomedical Technologies, Agenzia Nazionale per le Nuove Tecnologie, l’Energia e lo Sviluppo Economico Sostenibile (ENEA), 00123 Rome, Italy; flavia.novelli@enea.it (F.N.); simona.leonardi@enea.it (S.L.); caterina.merla@enea.it (C.M.); paola.giardullo@enea.it (P.G.); luca.march46@gmail.com (L.M.); annasaran60@gmail.com (A.S.); simonetta.pazzaglia@enea.it (S.P.); 2Department of Woman and Child Health and Public Health, Fondazione Policlinico Universitario Agostino Gemelli, Istituto di Ricovero e Cura a Carattere Scientifico (IRCCS), 00168 Rome, Italy; gabriele.babini@guest.policlinicogemelli.it; 3Department of Biological and Medical Sciences, Oxford Brookes University (OBU), Oxford OX3 0BP, UK; mkadhim@brookes.ac.uk; 4Radiation and Environmental Science Centre, Technological University Dublin, D02 HW71 Dublin, Ireland; damine.traynor@tudublin.ie (D.T.); dinesh.medipally@tudublin.ie (D.K.R.M.); aidan.meade@tudublin.ie (A.D.M.); fiona.lyng@tudublin.ie (F.M.L.); 5Helmholtz Zentrum München, German Research Center for Environmental Health GmbH (HMGU), Institute of Radiation Biology, D-85764 Neuherberg, Germany; soile.tapio@helmholtz-muenchen.de; 6Department of Agricultural and Forestry Sciences, Università della Tuscia, 01100 Viterbo, Italy; 7Department of Radiation Physics, Guglielmo Marconi University, 00193 Rome, Italy

**Keywords:** miRNome, Raman spectroscopy, abscopal effect, cardiac fibrosis, miR-1, mir-133a

## Abstract

**Simple Summary:**

Radiation exposure has been linked to non-cancer effects such as heart disease. This study aimed to investigate radiation-induced heart disease in mice where the radiation exposure was either administered to the whole body or only to the bottom third of the body (partial body). Radiation damage was found in the hearts of mice following both whole-body and partial-body exposure. MiRNAs released from directly irradiated skeletal muscle cells in vitro were shown to result in damaging effects in unirradiated ventricular cardiac cells. This study suggests that a partial-body exposure to radiation should be thought of as a systemic effect rather than only an effect on the exposed tissue.

**Abstract:**

Recent reports have shown a link between radiation exposure and non-cancer diseases such as radiation-induced heart disease (RIHD). Radiation exposures are often inhomogeneous, and out-of-target effects have been studied in terms of cancer risk, but very few studies have been carried out for non-cancer diseases. Here, the role of miRNAs in the pathogenesis of RIHD was investigated. C57Bl/6J female mice were whole- (WBI) or partial-body-irradiated (PBI) with 2 Gy of X-rays or sham-irradiated (SI). In PBI exposure, the lower third of the mouse body was irradiated, while the upper two-thirds were shielded. From all groups, hearts were collected 15 days or 6 months post-irradiation. The MiRNome analysis at 15 days post-irradiation showed that miRNAs, belonging to the myomiR family, were highly differentially expressed in WBI and PBI mouse hearts compared with SI hearts. Raman spectral data collected 15 days and 6 months post-irradiation showed biochemical differences among SI, WBI and PBI mouse hearts. Fibrosis in WBI and PBI mouse hearts, indicated by the increased deposition of collagen and the overexpression of genes involved in myofibroblast activation, was found 6 months post-irradiation. Using an in vitro co-culture system, involving directly irradiated skeletal muscle and unirradiated ventricular cardiac human cells, we propose the role of miR-1/133a as mediators of the abscopal response, suggesting that miRNA-based strategies could be relevant for limiting tissue-dependent reactions in non-directly irradiated tissues.

## 1. Introduction

A causal association between radiation exposure and various types of non-malignant diseases, especially cardiovascular diseases, has been extensively reported [1,2]. The risks of radiation-induced heart disease (RIHD) were described in therapeutically exposed cohorts of patients with thoracic tumors in which a high incidence of long-term complications (i.e., pericarditis, cardiomyopathy, coronary artery disease, valvular heart disease, conduction abnormalities and myocardial fibrosis) increased the risk of heart-disease-related mortality [3]. Patients who received post-mastectomy radiotherapy for left-sided breast cancer were at a 2–3-time higher risk of developing cardiovascular disease, due to the apex and anterior wall of the heart being exposed to doses of radiation between 1 and 5 Gy on average [4]. Nevertheless, at a median follow-up of 8 years, the risk of RIHD was not associated with the laterality of the irradiated breast [5]. An increased risk for cardiovascular diseases, especially stroke and heart attack, also resulted from epidemiological studies of A-bomb survivors [6] as well as occupationally exposed workers [7]. At these moderate doses (0.5–5 Gy), the mechanisms of action seem to especially involve atherosclerosis or vascular injury. Many animal studies support such evidence, indicating increased oxidative stress and the promotion of inflammation as possible mechanisms by which radiation promotes atherogenesis [8]. Cardiac fibrosis and hypertrophy were also observed after the irradiation of mice with 0.5 Gy of protons or 0.15 Gy of ^56^Fe ions, at late time points [9].

In the cardiovascular system, miRNAs control basic functions in virtually all cell types relevant to the cardiovascular system (such as endothelial cells, cardiac muscle, smooth muscle, inflammatory cells and fibroblasts); therefore, they are directly involved in the pathophysiology of many cardiovascular diseases [10]. Furthermore, they have essential roles in an effective cellular response to radiation exposure and major changes in systemic circulating miRNA profiles, and persistent alterations in local miRNA profiles in irradiated tissue have been reported [11]. In this context, the role of miRNAs in the pathogenesis of RIHD assumes particular importance in cases of inhomogeneous exposure, when the heart is out of the irradiation field. The presence of miRNAs in exosomes has, in fact, suggested that circulating miRNAs could have a role in cell-to-cell communication [12]. This would further suggest that miRNAs are selectively targeted for secretion in one cell and taken up by a distant, target cell, possibly in order to regulate gene expression.

The out-of-target effects of radiation are major concerns both for radiotherapy and radiation protection. In the past decades, their relevance has mainly been explored in terms of carcinogenic risk, and many in vivo experimental systems have provided evidence of out-of-target DNA-damage responses [13,14,15,16]. However, out-of-target effects are only beginning to be explored for non-cancer diseases. Recently, we reported that the effects of a 2 Gy partial-body irradiation (PBI) on the hippocampus, placed outside the radiation field, were nearly identical to those observed after whole-body irradiation (WBI) in terms of changes in miRNAs, proteins and rate of neurogenesis, thus providing a proof of principle of the existence of out-of-target radiation response in the hippocampus of conventional mice [17].

Here, with the same experimental strategy of WBI or PBI previously adopted, we focused on the heart with the purpose of: (*i*) understanding whether detrimental health consequences of a partial-body exposure can be considered a systemic effect rather than a tissue-specific effect; (*ii*) profiling the expression of miRNAs in directly irradiated and shielded hearts; (*iii*) discriminating the role of miRNAs as sensors, mediators or effectors of out-of-field effects.

To this aim, groups of C57Bl/6J female mice were WBI or PBI with 2 Gy of X-rays or sham-irradiated (SI). In PBI exposure, the lower third of the mouse body was irradiated, while the upper two-thirds were shielded. From all groups, hearts were collected 15 days or 6 months post-irradiation. A full mapping of the miRNA sequences via Next-Generation Sequencing (NGS) analysis was performed 15 days post-irradiation, followed by bioinformatics analyses. We identified a subset of miRNAs belonging to the myomiR family (miR-1, miR-133a, miR-133b, miR-206, miR-208a, miR-208b, miR-499, miR-486) to be highly differentially expressed in both whole- or partial-body-irradiated mouse hearts compared with their sham-irradiated counterparts. Furthermore, the miR-1/133a cluster was found to be significantly down-regulated in a mechanistic in vitro study designed ad hoc to demonstrate the crosstalk between skeletal muscle and ventricular cardiac human cells, as a critical event responsible for out-of-target radiation response in the heart and its sequelae.

## 2. Materials and Methods

### 2.1. Mice Irradiation and Dosimetry

C57Bl/6J female mice were WBI or PBI (n = 19 for each experimental condition) with 2 Gy of X-rays (dose rate = 0.89 Gy/min) at 8 weeks of age using a Gilardoni CHF 320 G X-ray generator (Gilardoni, Mandello del Lario, Italy) operated at 250 kVp, 15 mA, with Half-Value Layer = 1.6 mm Cu (additional filtration of 2.0 mm Al and 0.5 mm Cu). An additional group of mice (n = 19) was sham-irradiated (SI). To partially irradiate the mouse body, we used individual lead parallelepipeds to shield the anterior two-thirds of the body, with the hindmost part directly exposed to radiation (Figure 1a). In this configuration, kidneys were excluded by the irradiation field and only gut, vascular endothelium, bone marrow and predominantly muscle tissue were directly exposed to radiation. To ensure that the out-of-field effects under study were not the result of photons crossing the lead shield or deflected in the cap through the irradiated tissues, a NE 2571 ionization chamber was set in the same position as the heart of the irradiated mice and inserted into the lead parallelepiped cap with the same characteristics of the shields used to partially protect the mice from being irradiated (Figure 1b). This was repeated with or without a phantom. The estimation of dose into the shield, resulting from the average of 10 measurements, was 7.6 mGy, corresponding to 0.38% of the total dose (2 Gy dose at 250 kVp), thus indicating the absence of significant dose contribution to the shielded heart (Table 1).

### 2.2. Tissue Collection and Processing

Mice were housed under conventional conditions with food and water available ad libitum and a 12:12 h light–dark schedule until the collection of heart tissue 15 days and 6 months post-irradiation.

Total RNA was extracted from the lower half of mouse hearts (n = 3 for each time point) obtained from SI, WBI or PBI mice using an miRNeasy kit (QIAGEN, Milan, Italy) according to the manufacturer’s instructions. RNA was quantified by optical density (OD) measurements using a NanoDrop spectrophotometer (Thermo Fisher Scientific Inc., Milan, Italy), and quantification was performed using the Agilent TapeStation 200 (Agilent Technologies, Inc., Santa Clara, CA, USA). Six months post-irradiation (n = 3 for each experimental group), hearts were fixed in 10% buffered formalin, paraffin-embedded, sectioned, and stained with hematoxylin and eosin (H&E) and Masson’s trichrome or OCT-embedded. To quantify collagen after Masson’s trichrome staining, tissue sections were imaged with NIS-Elements BR software version 4.00.05 (Nikon Instruments Europe B.V., Amsterdam, The Netherlands) at 20× magnification and analyzed with HistoQuest 2.0.2.0249 software (TissueGnostics, Vienna, Austria) for automatic color separation and quantification. The experimental time points and the analyzed endpoints after mice irradiation are summarized in Figure 1c.

### 2.3. Raman Spectroscopy

OCT-embedded heart tissue was sectioned (10 μm) using a cryostat. An XploRA confocal Raman instrument (HORIBA Jobin Yvon) was used for spectral acquisition. The manual calibration of the grating was carried out using the 520.7 cm^−1^ Raman line of crystalline silicon. Dark-current measurements and the recording of the substrate and optics signal were also performed for data correction. As source, a 532 nm laser of ~12 mW power was focused by a 100X objective (MPlanN, Olympus, NA = 0.9) onto the sample, and the resultant Raman signals were detected using a spectrograph with a 1200 g/mm grating coupled with a CCD. Raman spectra were acquired in the 400–1800 cm^−1^ region with an integration time of 30 s per spectrum and averaged over two accumulations. Multiple calibration spectra of 1,4-Bis(2-methylstyryl)benzene were recorded along with each sample acquisition. All spectra were subsequently wavenumber-calibrated using in-house developed procedures in Matlab v.9.3 (Mathworks Inc., Natick, MA, USA). The instrument-response correction was performed using the spectrum of NIST Standard Reference Material (SRM) No. 2242.

Spectra were recorded from the heart tissue from 5 individual mice per group (SI, WBI or PBI). All spectral processing procedures were conducted using Matlab (R2017a; Mathworks Inc.), along with in-house developed algorithms and procedures available within PLS Toolbox (v 8.0.2; Eigenvector Research Inc., Wenatchee, MA, USA). Briefly, spectra were imported, and the baseline was subtracted with a rubberband algorithm; the vector was normalized and smoothed using the Savitzky–Golay smoothing algorithm with a 7-point window and a 5th order polynomial. Subsequently, the corrected spectra were subjected to a principal component analysis (PCA). In brief, the PCA is a commonly used method for multivariate data reduction and visualization. It is an unsupervised method used to describe the variance in data sets by identifying a new set of orthogonal features, called principal components (PCs).

A classical least squares (CLS)-fitting analysis was performed as described previously [18,19,20] using reference spectra of pure molecular species, which were purchased from Sigma-Aldrich (Wicklow, Ireland). In brief, CLS is an exploratory method that aims to minimize the squared differences between the fit and the spectrum using a set of reference pure molecular spectra. It assumes that any complex spectrum, S, is the linear sum of contributions from spectra of pure components, a1, a2,…an, that contribute to the spectrum as follows [21]:S = a1C1 + a2C2+…+E
where C1, C2,…Cn, are the weights or concentrations assigned to each component spectrum. In the case of a Raman spectrum, not all contributing pure components are known. Therefore, E represents the error or residual matrix. A CLS analysis was performed on 340 spectra from SI heart tissue, 282 spectra from PBI heart tissue and 343 spectra from WBI heart tissue. Significance testing was performed using two-tailed Student’s *t*-tests.

### 2.4. miRNome Analysis by Next-Generation Sequencing (NGS) and Bioinformatics Data Analyses

Total RNA (500 ng) was converted into miRNA NGS libraries as previously described [22,23], sequenced on Illumina NextSeq 500 System and then subjected to quality controls. A Pairwise Differential Expression (DE) analysis of the Tags Per Million (TPM) was performed using the R platform (http://www.r-project.org (accessed on 28 November 2018)) and the open-source Bioconductor libraries. Data were filtered as described [17]. Only statistically significant miRNAs (*p*-value ≤ 0.05, fold-change ≥ 2 for up-regulation and ≤2 for down-regulation) were used for the gene/miRNA enrichment analysis with Cytoscape plug-ins “ClueGo” (version 2.1.7) and “CluePedia” (version 1.1.7) [24] with a validated miRTarBase SCORE > 0.6. The top 20 predicted target genes for each miRNA in the list were finally selected to identify the affected pathways and functions on the REACTOME database (https://reactome.org, accessed on 28 November 2018), considering a minimum number of genes into the pathway equal to 20 with a percentage not less than 40 for the comparison between WBI and SI mouse hearts and equal to 5 with a percentage not less than 4 for the comparison between PBI and SI mouse hearts.

### 2.5. Real-Time qPCR

Total RNA (2 μg) was reverse-transcribed using High-Capacity cDNA Reverse Transcription Kit (Applied Biosystems, Foster City, CA, USA), and qPCR was performed with StepOnePluse Real-Time PCR System (Applied Biosystems) using Power SYBR Green PCR Master Mix (Applied Biosystems). The oligonucleotide primers used to evaluate gene expression are provided in Appendix A. Reactions were performed in triplicate from each biological replicate. Relative gene expression was quantified using glyceraldehyde-3-phosphate dehydrogenase (Gapdh) as housekeeping gene.

The MiRNA analysis was performed using TaqMan miRNA Assay (Thermo Fisher Scientific) for hsa-miR-1a (Assay ID 002246), hsa-133a (Assay ID 002222) and for U6 snRNA (Assay Name 01973) as housekeeping. The DDCt quantitative method was used to normalize the expression of the reference gene and to calculate the relative expression levels of target genes.

### 2.6. Cell Lines

Primary human skeletal muscle (SkMCs) and ventricular cardiac fibroblasts (NHCF-Vs) cell lines were purchased from Lonza (Lonza BioWhittaker Inc., Basel, Switzerland). Briefly, SkMCs were routinely maintained in SkBM Basal Medium (Lonza) supplemented with 0.1% hEGF, 0.1% GA-1000, 0.1% Dexamethasone, 2% L-glutamine and 10% FBS, while NHCF-Vs were plated in FBM Basal Medium (Lonza) supplemented with 0.1% Insulin, 0.1% hFGF-B, 0.1% GA-1000 and 10% FBS.

For co-culture experiments, 1 µm pore-size inserts (Corning Incorporated, Durham, NC, USA) were used. Both cell lines were seeded in differentiating medium: DMEM/F12 (Gibco, Amarillo, TX, USA), supplemented with 2 mM L-glutamine, 1% Penicillin-Streptomycin and 10% filtered FBS. SkMCs were seeded at densities of 65,000 and 75,000 cells/insert for 3 and 6 days, respectively. NHCF-Vs were seeded to the bottom of a 6-well plate at densities of 100,000 cells/well. Twenty-four hours after seeding, SkMCs plated in the insert were exposed to a single dose of 2 or 6 Gy of X-rays or left untreated. Irradiation was performed using a Gilardoni CHF 320G X-ray generator (Gilardoni S.p.A., Mandello del Lario, Italy) operated at 250 kVp and 15 mA, with Half-Value Layer = 1.6 mm Cu (additional filtration of 2.0 mm Al and 0.5 mm Cu). Subsequently, irradiated and non-irradiated SkMCs were transferred to the corresponding companion chambers containing untreated NHCF-Vs. Three and six days post-IR, co-cultured cells were collected, and total RNA was extracted using an miRNeasy kit (QIAGEN, Milan, Italy) according to the manufacturer’s instructions.

## 3. Results

### 3.1. Differentially Expressed miRNAs in the Hearts of Whole- or Partial-Body-Irradiated Mice and Pathway Analysis

At 15 days post-irradiation, we first compared the levels of miRNAs expressed in WBI vs. SI mouse hearts, obtaining 208 differentially expressed miRNAs in total with *p*-value ≤ 0.05. Of these, 142 were upregulated and 66 downregulated (Appendix A). The pathway analysis revealed that, among several deregulated molecular pathways, the most significant functions affected by miRNAs deregulation in the WBI condition converged on calcium (Ca^2+^) signaling, transforming growth factor β (TGF-β) pathway, cell-cycle regulation, and insulin and fibroblast-growth-factor signaling (Figure 2).

By comparing the levels of miRNAs expressed in PBI vs. SI mouse hearts, we found a reduced number of statistically significant deregulated miRNAs, i.e., 39, all down-regulated.

To explore more in depth miRNAs deregulation in the two different irradiation conditions, we intersected the statistically significant miRNAs perturbed in WBI and PBI mouse hearts. As shown in Figure 3, the Venn diagram highlighted that all 39 miRNAs found after PBI were in common with those identified after WBI (Table 2), with no miRNAs being contra-regulated between the groups. The pathway analysis of commonly deregulated miRNAs highlighted the perturbation of regulatory pathways with a critical role for cardiac function (Figure 4). Of note, the deregulation of miR-1a, miR-499, miR-133, miR-223 and miR-155 converges on the activation of the PKA-mediated phosphorylation of CREB, whose activation is associated with two well-recognized radiation-induced heart diseases, i.e., cardiac hypertrophy and heart failure [25,26], both strictly related to the alteration in Ca^2+^ signaling. In addition, the miR-1/133a cluster has been shown to regulate cardiac fibrosis [27]. Notably, the most downregulated miRNAs belonged to the myomir family described as striated-muscle specific (miR-1, miR-133a, miR-133b, miR-206, miR-208a, miR-208b and miR-499) or muscle-enriched (miR-486). Their circulating levels have been proposed to be new biomarkers of physiological and pathological muscle processes [28].

### 3.2. Effect of In-Field or Out-of-Field Irradiation on the Biochemical Profile of Heart Tissue

The biochemical profile of heart tissue 15 days following WBI or PBI was investigated using Raman spectroscopy. As shown in Figure 5a, the PCA of the Raman spectral data showed overlapping but distinct clusters for the SI, PBI and WBI groups. The principal component loading indicated that nucleic acids (728, 780, 1340, 1480, 1580 cm^−1^), proteins (820, 850, 890, 1450, 1600, 1660, 1680 cm^−1^) and lipids (1300, 1400, 1420 cm^−1^) were among the differentiating spectral features (Figure 5b).

Similar to data obtained 15 days after irradiation, Raman spectral data 6 months post-irradiation showed overlapping but distinct clusters for the SI, PBI and WBI groups (Figure 6a). The spectral features shown in the PC loading (Figure 6b) indicated that, again, nucleic acids (728, 780, 1580 cm^−1^), proteins (820, 850, 1450, 1600, 1660 cm^−1^) and lipids (1300, 1420 cm^−1^) were among the differentiating features. The CLS-fitting analysis was also performed to estimate the relative fraction of reference spectra of DNA, actin and TGF-β within the tissue spectra. The results showed that actin was significantly decreased in the WBI and PBI groups compared with the SI group, whereas TGF-β was found to be significantly increased in the WBI and PBI groups compared with the SI group (Figure 6c).

### 3.3. Impact of Whole- and Partial-Body Irradiation on the Cardiac Tissue 6 Months after Exposure

Myocardial fibrosis is considered the major stage of RIHD, affecting post-radiation-therapy survivors who received high doses of radiation to the heart [29]. The accumulation of collagen in the myocardium interstitium is synthesized by fibroblasts after they are differentiated and activated in response to different cytokines and growth factors such as TGF-β [30]. As the TGF-β pathway and Ca^2+^ signaling were deregulated in both WBI and PBI mouse hearts, we evaluated, with a multiple approach, the presence of fibrosis in the cardiac tissue 6 months after WBI and PBI.

Firstly, we evaluated the presence of collagen in cardiac sections after Masson’s trichrome staining, as shown in Figure 7a–c. After quantification, we observed a statistically significantly increased deposition of collagen throughout the myocardium in the absence of cell death in WBI and PBI mice, compared with SI mice (Figure 7d). Secondly, we quantified, by qPCR, the mRNA expression levels of four genes commonly involved in myofibroblast activation, i.e., vimentin, fibronectin1, α-smooth muscle actin (α-SMA) and collagen 3A1. As shown in Figure 7e–h, all genes were found to be significantly over-expressed compared with SI mouse hearts, regardless of the irradiation condition. Altogether, our data indicate that PBI is able to activate RIHD similarly, in quality and in quantity, to WBI.

### 3.4. Mechanistic Study to Investigate the Propagation of Signals between Irradiated and Non-Irradiated Cells

In our experimental scheme of PBI, we shielded the anterior two-thirds of the body with only gut, vascular endothelium, bone marrow and predominantly muscle tissue exposed directly to radiation. As muscle injury induces the release of many secreted factors [31], such as circulating miRNAs [28], during muscle repair and regeneration, we hypothesized a crosstalk between skeletal muscle and ventricular cardiac human cells, probably mediated by miRNAs, as a possible explanation of RIHD after PBI.

In an attempt to investigate this hypothesis under a more simplified condition than in vivo, we took advantage of a co-culture system of differentiated human SkMCs and NHCF-Vs, in which an insert with a permeable membrane allowed the secreted soluble factors to diffuse in the absence of cell–cell contact.

Focusing our attention on the miR-1/133a cluster, we first evaluated their basal intracellular expression in SkMCs and NHCF-Vs. Although miR-1 and miR-133 constitute the original (canonical) myomiRs and are considered muscle specific because of their prevalence in skeletal and cardiac muscle [32], their expression levels were highly statistically significantly different between SkMCs and NHCF-Vs. In particular, miR-1 was 2.5-fold (*p* = 0.019) and miR-133a 14-fold (*p* < 0.0001) more expressed in SkMCs compared with NHCF-Vs (Appendix A).

Then, we evaluated the expression levels of both miRs in SkMCs after irradiation with two different radiation doses, i.e., 2 and 6 Gy, at different time points post-irradiation. In Figure 8a,b, the kinetics of the expression of both miRNAs clearly indicated a statistically significant decreasing trend compared with SI cells over time and in a dose-dependent manner, reaching their minimum expression level 6 days post-irradiation. Thereafter, we opted for this time point for further co-culture experiments. SkMCs were irradiated and co-cultured with NHCF-Vs (named bystander cells) according to the scheme summarized in Figure 8c; thus, the intracellular expression of miR-1 and miR-133a was evaluated and compared with that of directly irradiated NHCF-Vs (named RX cells). As shown in Figure 8d,e, a nearly identical trend of statistically significant decreasing expression levels of both miRNAs were found in bystander and irradiated NHCF-Vs. Accordingly, all genes indicating myofibroblast activation, direct targets of miR-1/133a clusters, were found to be significantly over-expressed compared with SI cells in both bystander and irradiated NHCF-Vs with the highest radiation dose (6 Gy). Similarly, a clear trend of increase was observed after irradiation with the 2 Gy dose (Figure 8f,i).

## 4. Discussion

The use of radiation in medicine is an essential tool underlying advanced technologies such as radiotherapy, diagnostic radiobiology and nuclear medicine, all with a strong impact in the numerous areas of medical specialization. The benefits of using them in medical practice are indisputable, but their intrinsic properties are known to cause potential undesirable effects. It follows that an increasingly in-depth knowledge of the mechanisms that govern the interaction of radiation with biological matter is fundamental for a correct informed judgment on the risk/benefit ratio. An increased risk of developing second malignancies after having radiation therapy has been reported, with incidence varying with respect to the age of the patient at the time of treatment, the dose of radiation and the specific area treated [33]. In the last decade, there has also been increasing evidence reporting a link between radiation exposure and the increase in non-cancer disease incidence, especially RIHD, both after radiotherapy [34,35] and in epidemiological cohorts exposed at moderate and low dose levels [36,37]. The use of animal models has clarified many of the mechanisms responsible for cardiac damage induced by high radiation doses [8]. Much more limited are the studies focused on evaluating low-dose radiation effects on the heart. Nevertheless, a clear indication that both low (0.2 Gy) and moderate (2 Gy) doses of radiation locally delivered to the heart of *ApoE^−/−^* mice are effective to increase fibrosis markers has been reported [38]. Furthermore, in the era of the new paradigm of radiobiology, it is currently recognized that the detrimental effects of ionizing radiation are not limited only to directly irradiated cells but also occur in distant cells [39], implying consequences both for cancer [40] and non-cancer diseases [17], as well documented through the use of in vivo experimental models of inhomogeneous exposure.

In this work, using a multi-omics approach, we showed that when the heart was distant from directly irradiated tissues, significant changes occurred in miRNA contents and biochemical profile compared with hearts obtained from SI mice. Although the number of perturbed miRNAs was 5-fold lower (39 vs. 208) than that found when the heart was directly irradiated, all miRNAs were in common between the PBI and WBI groups, and all were downregulated. Notably, comparing the most significant REACTOME pathways associated with the statistically significant miRNAs derived from the intersection between PBI and WBI mouse hearts (Figure 5) and those from directly irradiated hearts (Figure 3), important similarities can be recognized. One of the most relevant predicted pathways found to be deregulated in WBI mouse hearts was the phosphoinositide 3-kinase (PI3K) cascade. This pathway is the master regulator of insulin action in the heart, and its activation is strictly interconnected with Ca^2+^ signaling, also found to be highly deregulated. This interdependency, by balancing contractility with metabolic control, is crucial for cells of the cardiovascular system and is emerging to play key roles in disease development [41]. The transient-receptor-potential channels whose role is to mediate Ca^2+^ signaling in cardiac fibroblasts are considered sensors of many different pathophysiological events that stimulate cardiac fibrogenesis [42]. Furthermore, several studies have demonstrated that the PI3K/Akt signaling pathway is involved in regulating the occurrence, progression and pathological formation of cardiac fibrosis via the regulation of cell survival, apoptosis, growth and cardiac contractility [43]. Ca^2+^ signaling was clearly predicted to be deregulated also in hearts from PBI mice. Importantly, TGF-β signaling was among the deregulated predicted pathways in both irradiation conditions. The pathway analysis associated with the statistically significantly deregulated miRNAs in WBI and PBI mouse hearts suggested that canonical signaling pathways participating in cardiac fibrosis were activated 15 days after irradiation.

Using a PCA, it was shown that the biochemical profile of SI heart tissue was different from WBI and PBI heart tissues both 15 days and 6 months post-irradiation. Distinct clusters for each group were observed, although these were overlapping due to the complexity of the Raman data and because the unsupervised PCA could not identify subtle differences between groups. Despite this, interestingly, as shown in the PC loadings (Figure 5b and Figure 6b), the same spectral features were responsible for the discrimination between the different groups at each timepoint, namely, nucleic acids (728, 780, 1580 cm^−1^), proteins (820, 850, 1450, 1600, 1660 cm^−1^) and lipids (1300, 1420 cm^−1^). As it is not possible to obtain information about specific proteins or molecules from Raman spectra of cells or tissues, a CLS analysis was used to look at components that were assumed to be dysregulated based on the miRNA results. Significant differences in TGFβ and actin were found for WBI heart tissue.

In agreement with the omics data, the statistically significant increase in collagen deposition in cardiac sections of both WBI and PBI mice, supported by the increased expression of genes known to be involved in myofibroblast activation, clearly demonstrated that the impact of WBI and PBI on cardiac tissue is similar and converges on the onset of radiation-induced myocardial fibrosis.

With a more and more increased understanding of mechanisms involved in the cellular responses to ionizing-radiation exposure, miRNAs are now recognized as potential biomarkers, radiosensitizers, therapeutical elements and also mediators of damage signals responsible for important consequences of irradiation such as abscopal effects [44,45].

From our miRNome data, the most significant downregulated miRNAs belonged to the family of circulating myomiR, a subset of tissue-specific miRNAs described as striated-muscle specific (miR-1, miR-133a, miR-133b, miR-206, miR-208a, miR-208b, and miR-499) or muscle-enriched (miR-486); of these, miR-206 and miR-133b are specific to skeletal muscle, and miR-208a is cardiac-muscle specific [28].

Numerous studies have reported the clear role of the downregulation of miR-1/133 clusters in cardiac fibrosis and remodeling [46]. In rats with chronic heart failure, a miR-133a mimic and miR-133a overexpression significantly decreased fibrosis by inhibiting serine/threonine kinase Akt [47]. In the study by Muraoka et al. [48], miR-133 overexpression inhibited the expression of numerous genes in fibroblasts, also activating the cardiac gene program. MiR-1 downregulation was reported in failing hearts, in the tissues of patients with dilated cardiomyopathy and aortic stenosis and in hearts with ischemic cardiomyopathy [49]. Of note, miR-1 is expressed in cardiac fibroblasts, which are among the major non-muscle cell types in the myocardium and are responsible for cardiac fibrosis under pathological conditions. The results obtained using an in vivo myocardial-fibrosis model, together with data from in vitro experiments in which cardiac fibroblasts were analyzed under pathologic stimulations by Ang II or TGFβ, demonstrated the down-regulation of miR-1, thus pointing to this miRNA as a player in the regulation of cardiac fibroblast function and signaling [50].

Since after cell injury miRNAs can be released from cells in an active or passive mode, we hypothesized that miR-1/133a circulating levels, secreted by injured muscle cells during the PBI setup, may be responsible for the RIHD onset observed in the out-of-target hearts. To this aim, we opted for an in vitro strategy, co-culturing a human skeletal muscle cell line, SkMCs, with ventricular cardiac fibroblasts (NHCF-Vs) without cell contact but allowing soluble factors to diffuse through an insert. With this simplified methodology, we were able to demonstrate the down-regulation of miR-1 and miR-133a in directly irradiated SkMCs but also in bystander-irradiated NHCF-Vs, with a consequent increased expression of genes involved in myofibroblast activation in both cell lines.

The lack of in vivo evidence that circulating miRNAs are directly involved in the RIHD onset of PBI mice represents a limitation of our study. However, a miRNome analysis of hippocampi obtained from the same PBI mice revealed 25 differentially expressed miRNAs compared to SI mice [17]. Of these, 4/25 belonged to the myomiR family, thus suggesting that these circulating miRNAs could have the potential to induce tissue-independent reactions in shielded tissues.

Furthermore, in another recent manuscript [51], we performed an NGS-based miRNome analysis of plasma exosomes from mice irradiated using the same experimental conditions and with the same radiation doses reported here (SI, PBI and WBI) 24 h post-irradiation. Compared with SI mice, we found 57 significantly differentially expressed miRNAs in plasma exosomes from WBI mice, 13 from PBI mice and only 5 in common between the two different irradiation conditions. Mir-1 was only differentially expressed among miRNAs in plasma-derived exosomes from WBI mice, while miR-133 was not detected due to its prevalence in the non-exosomal component [52]. It is important to consider that the release of miRNAs from radiation-injured tissues could be very different depending on the different time points analyzed (24 h vs. 15 days) thus reflecting acute and delayed tissue injury.

An induced abscopal response in non-targeted mouse hearts was described by Aravindan et al. 24 h after irradiation limited to the lower abdomen with clinically relevant radiation doses [53]. With a transcriptional profiling approach, the onset of NF-kB signal transduction and subsequent NF-kB activation was found in non-targeted tissues. Due to the role of NF-κB activity in the development of cardiac fibrosis [54], this result supports our conclusions, which go beyond these pre-existing data by demonstrating, for the first time, the onset of cardiac fibrosis in an in vivo experimental mouse model of PBI.

The existence of out-of-target radiation response in the heart and hippocampus [17] of PBI mice contributes to the idea that detrimental health consequences of an inhomogeneous exposure should be considered a systemic effect rather than a tissue-specific effect. Furthermore, many perturbed miRNAs and predicted target pathways were in common among the analyzed organs, indicating that miRNAs have key roles in this phenomenon and deserving further investigations as possible early biomarkers and/or mitigators of pathology for radiation-induced changes leading to non-cancer diseases.

## 5. Conclusions

In conclusion, here, we elucidated the molecular mechanisms through which the radiation exposure of distant tissues may cause non-cancer tissue reactions in the heart also after irradiation with a moderate dose of ionizing radiation. Similar to the effects induced by direct irradiation, we also found RIHD onset in the hearts of PBI mice. Mir-1 and miR-133a, members of the myomiR family, were highly downregulated after the NGS analysis carried out 15 days after irradiation in both WBI and PBI hearts and bioinformatics predicted the activation of signaling pathways involved in myofibroblast activation. This result was confirmed 6 months after irradiation by analyzing WBI and PBI hearts with a multiple approach. Finally, we propose the potential role of miR-1/133a as mediators of the abscopal response possibly through their secretion from radiation-injured distant muscle cells.

Our results are relevant in radioprotection and can lead to further investigations aimed at limiting specific tissue-dependent reactions in non-directly irradiated tissues.

## Figures and Tables

**Figure 1 cancers-14-03463-f001:**
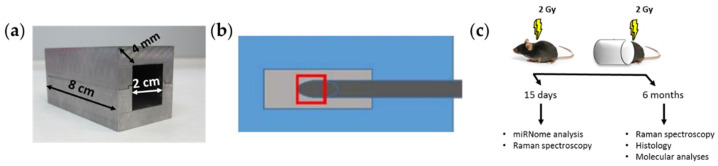
(**a**) Lead parallelepiped to shield the anterior two-thirds of the mouse body. (**b**) Position of the NE 2571 ionization chamber into the cap to estimate the scatter dose to the shielded heart. (**c**) Workflow of experimental endpoints after mice irradiation.

**Figure 2 cancers-14-03463-f002:**
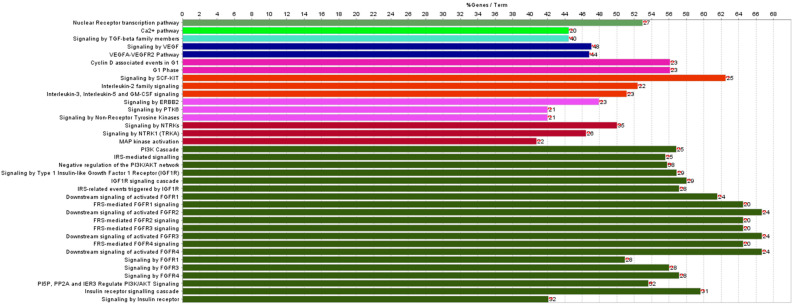
Histogram showing the most significant REACTOME pathways associated with the statistically significant miRNAs (after genes enrichment) altered in WBI vs. SI mouse hearts (listed in Appendix A). The single line represents specific gene ontology (GO) terms obtained from the ClueGo analysis. The colors indicates specific clusters of the differentially expressed genes between PBI and WBI samples. Red asterisks refer to significance in the percentage of pathway deregulation (* *p* < 0.05, ** *p* < 0.01).

**Figure 3 cancers-14-03463-f003:**
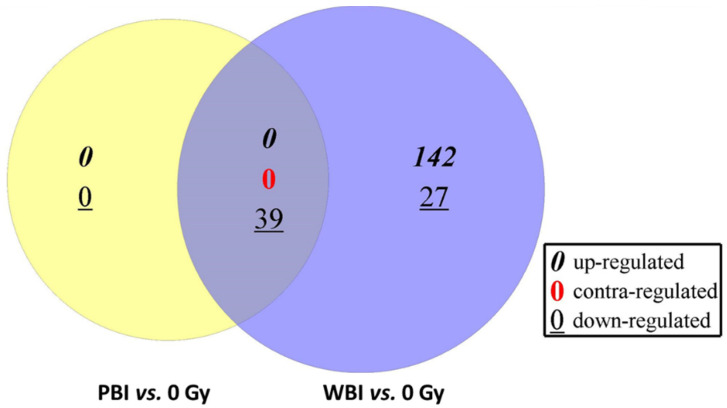
Venn diagram showing overlap between the statistically significant miRNAs perturbed in in PBI (**left** side) and WBI mouse hearts (**right** side). All miRNAs in common were down-regulated.

**Figure 4 cancers-14-03463-f004:**
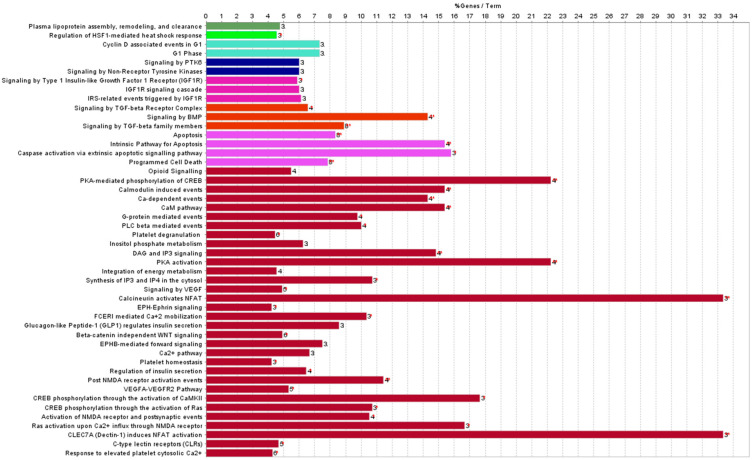
Histogram showing the most significant REACTOME pathways associated with the statistically significant miRNAs in common between PBI and WBI mouse hearts (listed in Table 2). The single line represents specific gene ontology (GO) terms obtained from the ClueGo analysis. The colors indicate specific clusters of the differentially expressed genes between PBI and WBI samples. Red asterisks refer to significance in the percentage of pathway deregulation (* *p* < 0.05, ** *p* < 0.01).

**Figure 5 cancers-14-03463-f005:**
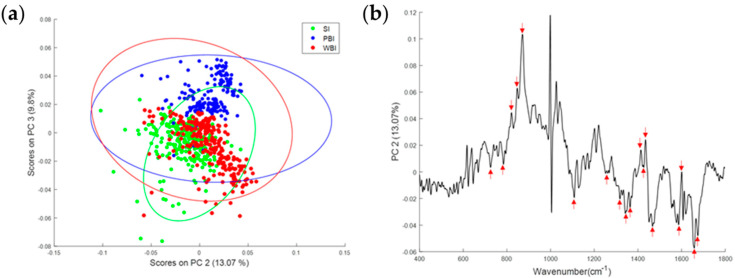
(**a**) PCA scatterplot of Raman spectral data from SI (green), PBI (blue) and WBI (red) mice 15 days post-irradiation. (**b**) PC loading from PCA of Raman spectral data from SI, PBI and WBI mice 15 days post-irradiation showing spectral features responsible for the separation between the groups (red arrows).

**Figure 6 cancers-14-03463-f006:**
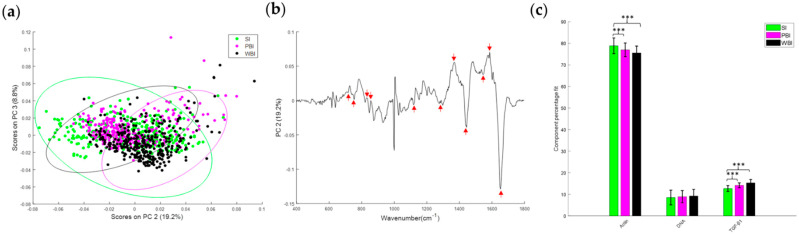
(**a**) PCA scatterplot of Raman spectral data from SI (green), PBI (magenta) and WBI (black) mice 6 months post-irradiation. (**b**) PC loading from PCA of Raman spectral data from SI, PBI and WBI mice 6 months post-irradiation showing the spectral features responsible for the separation among the groups (red arrows). (**c**) Relative weightings of actin, DNA and TGF-β from least squares fitting of Raman spectra from SI, PBI and WBI groups 6 months post-irradiation. Error bars represent the standard error. *** *p* ≤ 0.001.

**Figure 7 cancers-14-03463-f007:**
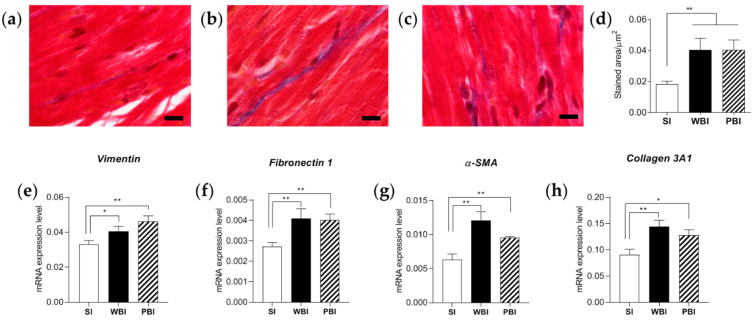
(**a**–**c**) Representative images of Masson’s trichrome staining of cardiac sections obtained from SI (**a**), WBI (**b**) and PBI (**c**) mice 6 months after exposure, in which collagen is stained blue. (**d**) Collagen-positive area quantified using HistoQuest 2.0.2.0249 software. Evaluation of *vimentin* (**e**), *fibronectin 1* (**f**), *α-SMA* (**g**) and *collagen 3A1* (**h**) expression levels, determined with qPCR analysis, 6 months after exposure. Each dataset represents the mean ± SEM of three independent biological replicates. * *p* < 0.05; ** *p* < 0.01 (Student’s *t*-test). Bars = 10 μm.

**Figure 8 cancers-14-03463-f008:**
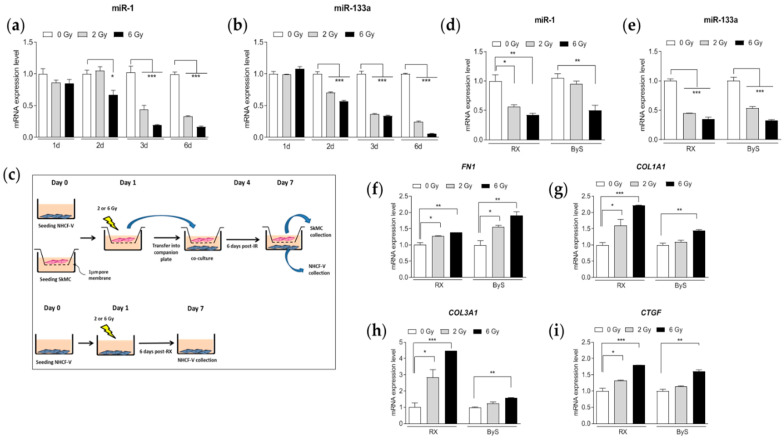
Intracellular levels of miR-1 (**a**) and miR-133a (**b**) in SkMCs after irradiation with 2 and 6 Gy, at different time points post-irradiation. (**c**) Schematic representation of the co-culture system. Intracellular levels of miR-1 (**d**) and miR-133a (**e**) in bystander and irradiated NHCF-Vs. Evaluation of *fibronectin 1* (*FN1* (**f**)), *collagen 1A* (*COL1A1* (**g**)), *collagen 3A1* (*COL3A1* (**h**)) and *connective tissue growth factor* (*CTGF* (**i**)) expression levels in bystander and irradiated NHCF-Vs. Each dataset represents the mean ± SEM of three independent biological replicates. Values obtained with sham-irradiated cells (0 Gy) were taken as 1. * *p* < 0.05; ** *p* < 0.01; *** *p* < 0.001; (Student’s *t*-test). RX = irradiated cells. ByS = bystander cells.

**Table 1 cancers-14-03463-t001:** Summary of the experimental measurements with the NE 2571 ionization chamber. Values of scatter doses to the heart at 60, 100, 200 and 250 kVp.

X-ray Quality Code	HLV/mm Cu	E/KeV	HV/V	I/mA	Dose (Gy)	Scatter Dose (mGy)
**H-60**	0.082	38	64.9	45	2	2.913
**H-100**	0.29	57.3	102.7	40	2	6.476
**H-200**	1.61	99.3	201.5	20	2	9.364
**H-250**	2.44	121.5	250	15	2	7.614

HLV = Half-Value layer; E = energy; HV = high voltage; I = current.

**Table 2 cancers-14-03463-t002:** List of statistically significant miRNAs in common between PBI and WBI mouse hearts. These 39 miRNAs are the ones of the Venn diagram in Figure 3 obtained by intersecting deregulated miRNAs in PBI and WBI.

miRNA	LogFC ^a^	miRNA	LogFC
mmu-miR-208a-3p	−15,223	mmu-miR-214-3p	−5140
mmu-miR-208a-5p	−12,008	mmu-miR-378b	−4834
mmu-miR-133a-5p	−10,531	mmu-miR-378a-5p	−4785
mmu-miR-133a-3p	−10,067	mmu-miR-199a-5p	−4757
mmu-miR-1a-3p	−9184	mmu-miR-155-5p	−4727
mmu-miR-1a-1-5p	−10,678	mmu-miR-199a-3p	−4441
mmu-miR-133b-3p	−9139	mmu-miR-199b-3p	−4357
mmu-miR-499-5p	−8787	mmu-miR-486-3p	−4422
mmu-miR-499-3p	−10,697	mmu-miR-224-5p	−4046
mmu-miR-1a-2-5p	−8451	mmu-miR-122-5p	−3708
mmu-miR-208b-3p	−6388	mmu-miR-223-3p	−3185
mmu-miR-10a-5p	−6035	mmu-miR-143-5p	−3091
mmu-miR-378d	−5986	mmu-miR-322-5p	−2994
mmu-miR-10a-3p	−6466	mmu-miR-450b-3p	−2982
mmu-miR-199b-5p	−5793	mmu-miR-145a-3p	−2846
mmu-miR-10b-5p	−5396	mmu-miR-490-3p	−2716
mmu-miR-486-5p	−5366	mmu-miR-126a-5p	−2577
mmu-miR-3107-5p	−5354	mmu-miR-126a-3p	−2577
mmu-miR-378c	−5239	mmu-miR-27a-3p	−2564
mmu-miR-378a-3p	−4946		

^a^ FC = fold change, ratio between miRNA expression in PBI and WBI.

## Data Availability

The data presented in this study are available upon request from the corresponding authors.

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
