# Peer review of "MiRNA-Mediated Fibrosis in the Out-of-Target Heart following Partial-Body Irradiation"

_cancers, 2022, doi:10.3390/cancers14143463_

Round 1

Reviewer 1 Report

The study described in this manuscript used a female mouse model to examine how direct radiation effects on the heart in whole body exposure to 2 Gy X-rays compare with effects in the heart of mice in which only the bottom one third of the mouse was exposed to 2 Gy. In addition, in a co-culture model, irradiated human skeletal myocytes induced the expression of fibrosis-related genes in unirradiated human cardiac fibroblasts. The authors conclude that, even if the heart remains outside the radiation field, mediators (such as miRNAs) may be released from irradiated tissues and cause cardiac fibrosis.

A major concern regarding this manuscript is that the conclusions are too strong.

-          The authors state that they observe cardiac radiation fibrosis. However, the in vivo data provided do not provide strong evidence that cardiac fibrosis was seen. Masson Trichrome staining was not quantified. Fibrosis-related mediators such as vimentin were only measured on the RNA level and not on the protein level. Moreover, prior publications on cardiac effects of low-dose whole body irradiation or local heart irradiation in mouse models do not report cardiac fibrosis in those models.

-          This manuscript does not provide in vivo evidence for a role of circulating miRNAs in cardiac changes in the PBI model. Extracellular miRNAs were not examined in the cell culture model. Lastly, we cannot be certain that miRNAs in the tissue at 15 days after irradiation contribute to the development of fibrosis at 6 months.

Individual comments address the above and other concerns.

1.      Summary:

a.      Please specify which cells were used for the in vitro studies.

b.      Typo lines 31-32: thought of as a …

2.      Abstract:

a.      Please specify what part of the animal was exposed in PBI.

b.      Please specify which cells were used for the in vitro studies.

3.      Introduction:

a.      First paragraph: The cardiovascular effect seen in patients who receive 1-5 Gy is mostly ischemic heart disease due to accelerated atherosclerosis or vascular injury. Cardiac fibrosis is randomly seen in patients after such low doses. It is important to put the background information in this paragraph in the right context of what is studied in the mice. Therefore, please explain in the text.

b.      Line 90: Please specify what part of the mice was irradiated in the partial-body irradiation model.

c.       Line 98: Please indicated that these are human skeletal myocytes.

d.      Line 99: In the methods section, it is stated that human cardiac fibroblasts were studied, not cardiomyocytes. Please correct.

4.      Materials and Methods:

a.      Paragraph 2.1: Please explain in this paragraph what part of the mouse was exposed in the PBI model, and which internal organs. Were the kidneys in the radiation field?

b.      Paragraph 2.1 or Table 1: Please include the dose rate.

c.       Table 1: Please add notes to spell out the abbreviations used in this table.

d.      Table 1: It seems that dosimetry readings were taken with the ionization chamber placed in air. In that case, one cannot say that the readings indicate dose to the heart (dose to the heart should have been investigated using mouse phantoms). Please change the wording to something like “external dose to the chest”.

e.      Paragraph 2.2: Which parts of the heart were used to extract RNA? Please specify.

f.        Paragraphs 2.2 and 2.3: There were 6 animals per group (3 animals per time point). It is not entirely clear how 3 hearts per group were used to collect RNA, perform histology, and how the authors were able to include 5 mice per group for Raman spectroscopy. Raman was performed at 2 time points, so 3 mice per group seems the maximum. Please clarify in the text.

5.      Results:

a.      Paragraph 3.1: Please specify at which time point these data were collected.

b.      Table 2 is not entirely clear. What is indicated by the Log Fold Change? Fold change of PBI compared to WBI? Also, how can we tell from table 2 that these miRNAs were regulated in common between PBI and WBI?

c.       Lines 263 and 273: Please change overlapped into overlapping.

d.      Figure 6C: Please explain what statistical test was used on these data. The differences between the groups are very small. How can they result in a very significant p-value <0.001, especially since the experimental groups were so small?

e.      Lines 289-290: Please indicate that myocardial fibrosis occurs in radiotherapy survivors who received high doses of radiation to the heart.

f.        Lines 301-304: In light of the above major concerns, the statement in lines 301-304 is too strong.

g.      Figure 7a-c: Please quantify the amount of collagen in the whole heart of each animal and include those data.

6.      Discussion:

a.      Please include a section to describe the results regarding cardiac radiation fibrosis after 2 Gy WBI, PBI or locally to the heart, in prior publications and discuss how the results in the current manuscript compare with those prior data.

b.      Please explain that a major limitation of this study is a lack of in vivo evidence of circulating miRNAs, and that miRNAs were also not measured in the culture media of the co-culture system.

c.       Please explain that examination of fibrosis-related mediators only on the gene expression level is a limitation of the study and is not direct evidence of fibrosis.

d.      Typo line 453: these pre-existing data.

e.      Lines 453-455: This conclusion is too strong. The study does not provide direct evidence for cardiac fibrosis nor does it elucidate the origin and mediators involved in vivo.

7.      Conclusions:

a.      Lines 473-476: Partial body exposures as performed in this study are not the norm for radiation therapy. Radiation therapy is most of the time much more focused to a smaller part of the body. Therefore, the mouse model is not relevant for human exposures during radiation therapy. The mouse model seems more relevant for WBI or PBI that may occur in a radiation accident. Please reword.

Author Response

The Authors thank this reviewer for his/her helpful comments, which have allowed us to improve the manuscript.

  1. Summary
  • Please specify which cells were used for the in vitro studies

 This information has been added in the revised manuscript.

  • Typo lines 31-32: thought of as a …

The typo has been corrected.

  1. Abstract
  • Please specify what part of the animal was exposed in PBI. Please specify which cells were used for the in vitro studies.

According to the requests, we have specified what part of the animal was exposed in PBI and which cells were used for the in vitro studies.

  1. Introduction
  • First paragraph: The cardiovascular effect seen in patients who receive 1-5 Gy is mostly ischemic heart disease due to accelerated atherosclerosis or vascular injury. Cardiac fibrosis is randomly seen in patients after such low doses. It is important to put the background information in this paragraph in the right context of what is studied in the mice. Therefore, please explain in the text.

We have added information in the first paragraph and, accordingly, references have been modified.

  • Line 90: Please specify what part of the mice was irradiated in the partial-body irradiation model. Line 98: Please indicated that these are human skeletal myocytes.

Line 99: In the methods section, it is stated that human cardiac fibroblasts were studied, not cardiomyocytes. Please correct.

We have specified what part of the animal was exposed in PBI and suggested modifications about cell lines have been addressed.

  1. Materials and Methods
  • Paragraph 2.1: Please explain in this paragraph what part of the mouse was exposed in the PBI model, and which internal organs. Were the kidneys in the radiation field?

We have specified that kidneys are not included in the irradiation field.

  • Paragraph 2.1 or Table 1: Please include the dose rate. Table 1: Please add notes to spell out the abbreviations used in this table.

We have included the dose rate (0.89 Gy/min) and modified the Table 1, spelling out the abbreviations.

  • Table 1: It seems that dosimetry readings were taken with the ionization chamber placed in air. In that case, one cannot say that the readings indicate dose to the heart (dose to the heart should have been investigated using mouse phantoms). Please change the wording to something like “external dose to the chest”.

Dosimetry has been carried out with or without a phantom; this information has been added in the revised manuscript.

  • Paragraph 2.2: Which parts of the heart were used to extract RNA? Please specify.

RNA has been extracted from the lower part of the heart (right and left ventricules); we have added this information in the text

  • Paragraphs 2.2 and 2.3: There were 6 animals per group (3 animals per time point). It is not entirely clear how 3 hearts per group were used to collect RNA, perform histology, and how the authors were able to include 5 mice per group for Raman spectroscopy. Raman was performed at 2 time points, so 3 mice per group seems the maximum. Please clarify in the text.

We thank the Reviewer for pointing out this error. The total number of mice enrolled in the study has been corrected.

  1. Results
  • Paragraph 3.1: Please specify at which time point these data were collected.

In paragraph 3.1, we have specified at which time point these data were collected.

  • Table 2 is not entirely clear. What is indicated by the Log Fold Change? Fold change of PBI compared to WBI? Also, how can we tell from table 2 that these miRNAs were regulated in common between PBI and WBI?

In Table 2, Log FC means the logarithmic value (in base 2) of the FC, and this is a standard metric in bioinformatics. The FC is the ratio between the miRNA expression in PBI and WBI, this point has been better detailed into the note of Table 2. This table simply reports the list of the 39 miRNAs obtained from the Venn diagram in Fig. 3. These miRNAs were obtained by intersecting the deregulated miRNAs in PBI and the ones in WBI. This detail has been added into the title of Table 2.

  • Lines 263 and 273: Please change overlapped into overlapping.

Overlapped has been changed to overlapping

  • Figure 6C: Please explain what statistical test was used on these data. The differences between the groups are very small. How can they result in a very significant p-value <0.001, especially since the experimental groups were so small?

Significance testing was performed using two-tailed Student t tests. This information has been added to the Methods section. CLS analysis was performed on 340 spectra from SI heart tissue, 282 spectra from PBI heart tissue and 343 spectra from WBI heart tissue.

Figure 6C has been updated in the revised manuscript as some errors were noticed when checking the significance testing.

  • Lines 289-290: Please indicate that myocardial fibrosis occurs in radiotherapy survivors who received high doses of radiation to the heart.

We have specified that myocardial fibrosis occurs in radiotherapy survivors who received high doses of radiation to the heart.

  • Lines 301-304: In light of the above major concerns, the statement in lines 301-304 is too strong.

Figure 7a-c: Please quantify the amount of collagen in the whole heart of each animal and include those data.

We quantified the collagen in the cardiac sections obtained from SI, WBI and PBI mice using the HistoQuest 2.0.2.0249 software, dedicated to the FACS-like analysis of samples stained with immunohistochemical or histochemical stains. These results (Fig. 7d) clearly indicate that the quantification is in line with our previous qualitative analysis. We thank this reviewer for his/her suggestion that improved our results and conclusions. In light of this result, we believe that the statement reported in line 331-332 of the revised version of the manuscript is consistent with the obtained data.

  1. Discussion:
  • Please include a section to describe the results regarding cardiac radiation fibrosis after 2 Gy WBI, PBI or locally to the heart, in prior publications and discuss how the results in the current manuscript compare with those prior data.

A paragraph has been added in the Discussion section to describe results regarding cardiac radiation fibrosis after local irradiation of mouse hearts with low and moderate dose of radiation.  

  • Please explain that a major limitation of this study is a lack of in vivo evidence of circulating miRNAs, and that miRNAs were also not measured in the culture media of the co-culture system.

We agree with the criticism raised by this Reviewer. Designing an in vivo experimental strategy to demonstrate a direct role of circulating miRNAs is not easily feasible and miRNAs were not measured in the co-culture system due to technical limitation (i.e., isolation and normalization in the medium). We toned down our conclusions, suggesting a possible role of myomiR as mediators of radiation damage supported by comparison with results obtained in our previous studies. The discussion section has been modified accordingly.

  • Please explain that examination of fibrosis-related mediators only on the gene expression level is a limitation of the study and is not direct evidence of fibrosis.

In light of quantitative results highlighting a statistically significant increase of collagen deposition in both WBI and PBI mouse hearts, we believe that a direct evidence of fibrosis has been proven.

  1. Conclusions:
  • Lines 473-476: Partial body exposures as performed in this study are not the norm for radiation therapy. Radiation therapy is most of the time much more focused to a smaller part of the body. Therefore, the mouse model is not relevant for human exposures during radiation therapy. The mouse model seems more relevant for WBI or PBI that may occur in a radiation accident. Please reword.

According to the request, the sentence has been reformulated.

Reviewer 2 Report

The authors adopted multi-omics methods to investigate radiation-induced heart disease in mice where the radiation exposure was either to the whole body or only to the bottom third of the body. There are several problems and suggestions:
1. In the 3.1 of RESULTS, please explain the meaning of the different colors in the histogram in Figure 2. If possible, please quote some references to illustrate pathways you find instead of just listing them in the text. In addition, the Venn plot
2. Please add a figure of workflow that should include the experimental steps and analysis steps.
3. In the 3.2 of RESULTS, please simply illustrate why the biochemical profiles of 15 days and 6 months were different. And give your assumptions about why some proteins were dysregulated and why PCA scatterplots showed overlapped but distinct clusters.
4. I recommend you try more bioinformatic analysis. In the discussion, you mentioned that in your recent manuscript, you performed an NGS-based miRNome analysis of plasma exosomes from mice irradiated using the same experimental conditions and with the same radiation doses reported here (SI, PBI, and WBI) at 24 h post-irradiation. If possible, you could make a combination of these data from 24h, 15 days, and 6 months and analyze the changes during a time course. Maybe in this way, you could find more insights.
5. In 3.3 of RESULTS, the legend of Figure 7 did not indicate “c”, please check carefully. And you’d better use arrows to indicate what was collagen.

Author Response

The Authors thank this reviewer for his/her helpful comments, which have allowed us to improve the manuscript.

  1. In the 3.1 of RESULTS, please explain the meaning of the different colors in the histogram in Figure 2. If possible, please quote some references to illustrate pathways you find instead of just listing them in the text. In addition, the Venn plot

The single line in Fig. 2 represents specific gene-ontology terms (GO) obtained from the CleuGo analysis. The color indicates specific clusters of differentially expressed genes between PBI and WBI samples. These details have been added into the caption of Fig. 2 and Fig. 4.

  1. Please add a figure of workflow that should include the experimental steps and analysis steps.

A workflow has been added in Figure 1 (panel c)

  1. In the 3.2 of RESULTS, please simply illustrate why the biochemical profiles of 15 days and 6 months were different. And give your assumptions about why some proteins were dysregulated and why PCA scatterplots showed overlapped but distinct clusters.

Thanks to the Reviewer for their feedback. We have included the following text to further explain these results in the Discussion section

“Using PCA, it was shown that the biochemical profile of SI heart tissue was different from WBI and PBI heart tissue at both 15 days and 6 months post irradiation. Distinct clusters for each group were observed although these were overlapping due to the complexity of the Raman data and because unsupervised PCA cannot identify subtle differences between groups. Despite this, interestingly, as shown in the PC loadings (figure 5b and 6b), the same spectral features were responsible for the discrimination between the different groups at each timepoint, nucleic acids (728, 780, 1580 cm-1), proteins (820, 850, 1450, 1600, 1660 cm-1) and lipids (1300, 1420 cm-1).  As it is not possible to obtain information about specific proteins or molecules from Raman spectra of cells or tissues, CLS analysis was used to look at components that were assumed to be dysregulated based on the miRNA results. Significant differences in TGFb and actin were found for the WBI heart tissue.”

  1. I recommend you try more bioinformatic analysis. In the discussion, you mentioned that in your recent manuscript, you performed an NGS-based miRNome analysis of plasma exosomes from mice irradiated using the same experimental conditions and with the same radiation doses reported here (SI, PBI, and WBI) at 24 h post-irradiation. If possible, you could make a combination of these data from 24h, 15 days, and 6 months and analyze the changes during a time course. Maybe in this way, you could find more insights.

NGS sequencing and then data normalization to perform bioinformatics analysis is a very delicate task and specific rules need to be respected to guarantee the robustness and reliability of the whole procedure and of the final results. The suggested mix of data (from this manuscript and from miRNome analysis of plasma exosomes), even if it seems very interesting it is not possible. The reason for this impediment is the used normalization procedures. Indeed for the exosome data the normalization procedure is different from the one used in this paper. The RNA of exosome samples were mailed to System Biosciences [SBI] Service [https://www.systembio.com/services/exosome-services/exo-ngs] as also reported in our recent manuscript (see S. Pazzaglia et al., Int. J. Mol. Sci. 2022), and their web-based service included library sequence quality control metrics, normalization of raw sequence read, data analysis for relative RNA abundance and identity and differential expression analysis. Therefore, the data normalization was performed by a service and we are not able to normalize those data in the same modality of the one in this paper. The service did normalization using closed algorithms and procedures to which we do not have access (no raw sequence are available). 

This point is unfortunately one of the current limitation of NGS-based analyses, which need to be performed for reliability using same normalization procedures for data to be compared.

Therefore, in the present situation, we are not able to mix these data sets in a reliable a rigorous fashion. This mixing procedure will only create false connections and interpretations. No further comments have been added into the revised manuscript concerning this point.

  1. In 3.3 of RESULTS, the legend of Figure 7 did not indicate “c”, please check carefully. And you’d better use arrows to indicate what was collagen.
    These suggestions have been addressed in the revised version of the manuscript. As in a standard Masson's Trichrome procedure collagen is stained blue, we have added this information in the figure legend.

.

Round 2

Reviewer 1 Report

Comments have been addressed

Reviewer 2 Report

The authors have answered my questions.